# Risk of COVID-19 transmission on long-haul flights: During the COVID-19 pandemic

**Jiyun Park[1], Gye jeong Yeom[ID][2]***

**1** Incheon Airport National Quarantine Station, Korea Centers for Disease Control and Prevention (KDCA), Incheon, Republic of Korea, **2** Department of Nursing Science, JEI University, Incheon, Republic of Korea

* salt42@hanmail.net

**Data Availability Statement:** All relevant data are within the paper and its Supporting Information file.

**Funding:** The author(s) received no specific funding for this work.

## Abstract

This study aimed to determine the possibility of COVID-19 transmission through in-flight contact during flights for many patients with confirmed COVID-19 during the COVID-19 pandemic and explore infection prevention and control (IPC) methods for overseas infectious diseases. A retrospective cohort study was conducted on flight with a large number of confirmed case among. Delhi- Incheon flights in 2020. This flight was selected to confirm transmission through close contact with the cabin, with a total of 14 confirmed cases out of 190 passengers (including 10 flight attendants). After confirming COVID-19 test results for those entering Korea, we conducted an epidemiological investigation on confirmed patients to determine their general characteristics and epidemiological relevance. We analyzed the epidemiological relevance, occupational information, incubation period, and COVID-19 variation and genotype among confirmed patients who were in close contact with confirmed cases, and analyzed the possibility of transmission according to the distance of close contact in the flight. One confirmed patient was found to be highly likely to be infected due to close contact with the cabin. However, it occurred within two rows, not within 1 meter. In addition, considering the aerodynamics in the cabin and local incidence rate, infection in an unspecified number of local people could not be excluded. It was analyzed that the reason for reducing infection from close contact on board for a long time in a flight with a large number of confirmed cases was the effective IPC method. In order to prevent overseas infectious diseases caused by flights, autonomous IPC management of airlines and passengers is necessary in addition to national quarantine management such as symptom screening before boarding, wearing passenger masks while boarding, food and beverage restrictions, disinfection of public spaces, distancing between passengers, close contact management after boarding, and self-quarantine.

## Introduction

Incheon Airport National Quarantine Station (IANQS) is implementing a preemptive response by establishing a close cooperative system with related agencies to prevent inflow of overseas infectious diseases into the country [1]. Due to the outbreak of pneumonia of

**Competing interests:** The authors have declared that no competing interests exist.

unknown cause in Wuhan, Hubei Province, China in December 2019, IANQS strengthened fever monitoring and quarantine procedures on January 3, 2020 and confirmed the first case of a patient infected by COVID-19 in Korea during the quarantine stage on January 19, 2020.

After the first patient with COVID-19 in Korea was confirmed at the IANQS, IANQS applied guidelines for responding to Middle East Respiratory Syndrome, Class 1 infectious disease, to actively monitor the range of close contacts in the cabin in three rows in front of and behind the confirmed patient's seat (within seven rows including the confirmed patient row) [2]. Subsequently, from November 2021, under the judgment of the public health doctor and epidemiologist, it was classified into two rows before and after the confirmed patient seat (within a total of five rows including the confirmed patient row) and monitored [3,4]. In previous studies related to in-flight cross-infection, the risk of infection was higher in front seats than in back seats and the possibility of transmission was high within the second row. Additionally, compared to community transmission, the risk of cross-infection on board was low due to the structural environment such as the aircraft airflow system and high efficiency partial air (HEPA) filters [3,5–8].

After the COVID-19 pandemic, research studies on risk factors of in-flight transmission and methods to reduce them became active [6,9–12]. Based on this, major airlines, International Air Transport Association (IATA), International Civil Aviation Organization (ICAO) started to wear masks and restrict food and beverage. Infection control guidelines such as social distancing and symptom screening before boarding were then published and applied [13]. After applying guidelines for in-flight infection prevention and control (IPC), the possibility of in-flight transmission has been found to be extremely low [12,14,15]. It is time to check the possibility of in-flight transmission and reduce social cost of close contact management according to changes in major infection control policies in Korea. When calculating direct and indirect social costs, including epidemiological investigation, management costs, and productivity loss of COVID-19 infected patients, a loss of at least 44 million won per person was estimated. Thus, it is necessary to control the amount of indiscriminate social distancing through targeted quarantine [16].

Therefore, this study aimed to determine the risk of in-flight transmission in an aircraft by conducting a retrospective cohort study focusing on long-distance flights (Delhi, India-Seoul, Korea), where many patients were confirmed with COVID-19 infection during the COVID-19 pandemic. By analyzing the risk by close contact distance in detail, this study intended to analyze the validity of domestic close contact classification criteria (2 rows front and back, total 5 rows including confirmed patient rows). By checking the IPC Protocol during the COVID pandemic, this study provides basic data for effective preemptive quarantine response and establishing a quarantine system at airport quarantine station.

## Methods

### Design

This was a retrospective cohort study of passengers on flights with many patients confirmed with COVID-19 infections from April 6, 2021 to May 5, 2021. During this period, quarantine measures for overseas arrivals were strengthened. Both Koreans and foreigners underwent real-time polymerase chain reaction (PCR) tests at the local community health center on the first day of arrival. Even if the result was negative, they would be required to perform home quarantine for two weeks. If symptoms appeared during the self-quarantine period after a negative result, a PCR test was conducted at the discretion of the local government epidemiological investigator. A PCR test was additionally conducted one day before release from self-quarantine. If the result was negative, self-quarantine was finally released.

## Participants

The total number of flight passengers analyzed in this study was 190, including 10 flight attendants (Indian nationals) and 180 passengers (2 Indian nationals and 178 Korean nationals). In this study, flight attendants were exempt from self-quarantine as they could not be followed up after entering the country. Therefore, they were excluded from the study. There were no confirmed patients shared in the International Health Regulation. Two foreign nationals who performed self-quarantine were included in this study. During the analysis period, a total of 14 people were confirmed to be infected by COVID-19. As a result of a close contact survey of 14 confirmed patients within 3 rows, 150 were found. Thus, the final study subject of this study was 164.

## Procedure

**Data collection.** This study collected data by adding a 14-day period as of April 21, 2021, the date of the last patient confirmed with COVID-19, in consideration of the longest incubation period of COVID-19 (1–14 days). Data used for this study included test results and basic epidemiological investigation of confirmed COVID-19 patients reported to the COVID-19 Information Management System (CIMS, https://covid19.kdca.go.kr), which is managed by the Korea Disease Control and Prevention Agency (KDCA) in accordance with the Infectious Disease Control and Prevention Act and the Quarantine Act. From CIMS, clinical symptoms characteristics, COVID-19 confirm date, COVID-19 epidemiological relevance, occupational information, COVID-19 variations, and genotype information of confirmed patients were then collected. If additional epidemiological investigations were needed, the lead author of this study, an epidemiologist, directly talked to the confirmed patient by phone to collect information. In accordance with Article 29, Paragraph 4 of the Quarantine Act, passenger reservation information, seat maps, and passenger lists for the relevant flight were requested and collected from the airline to identify close contacts. The data were accessed for research purposes on September 22th, 2023.

**Case definition and epidemiological investigations.** During this study period, the quarantine measure for all overseas arrivals, regardless of clinical symptoms, was to collect samples from the upper or lower respiratory tract for PCR testing on the day of entry or within one day at a local public health center. Accordingly, COVID-19 confirmed patients of this study were cases in which the COVID-19 gene (PCR) was detected in diagnostic tests conducted regardless of clinical features according to the definition of the COVID-19 community response guidelines [17].

Based on results of the basic epidemiological investigation registered in the COVID-19 information management system when COVID-19 was confirmed, results of the basic epidemiological investigation form were checked to identify age, gender, clinical symptoms, COVID-19 confirm date, underlying diseases, and detailed epidemiological relevance, initial clinical symptoms, clinical symptoms, drug use, occupational information, and accompanied entry travel status. In addition, by analyzing the list of arrivals and seat maps received through the airline, close contacts of 14 confirmed patients (3 rows front and back, total 7 rows including confirmed patient rows) were classified. According to the close contact classification criteria, in cases where there is a gap between aisles and areas, only the row immediately preceding the gap was classified. To check whether there was close contact on board, close contacts whose epidemiological relevance had already been confirmed were excluded. Cases with earlier COVID-19 confirmation days than the index case were also excluded. All patients confirmed in this study underwent additional variant testing. Even if they were confirmed to be close contacts, cases with different genotypes were excluded from infection due to in-flight transmission.

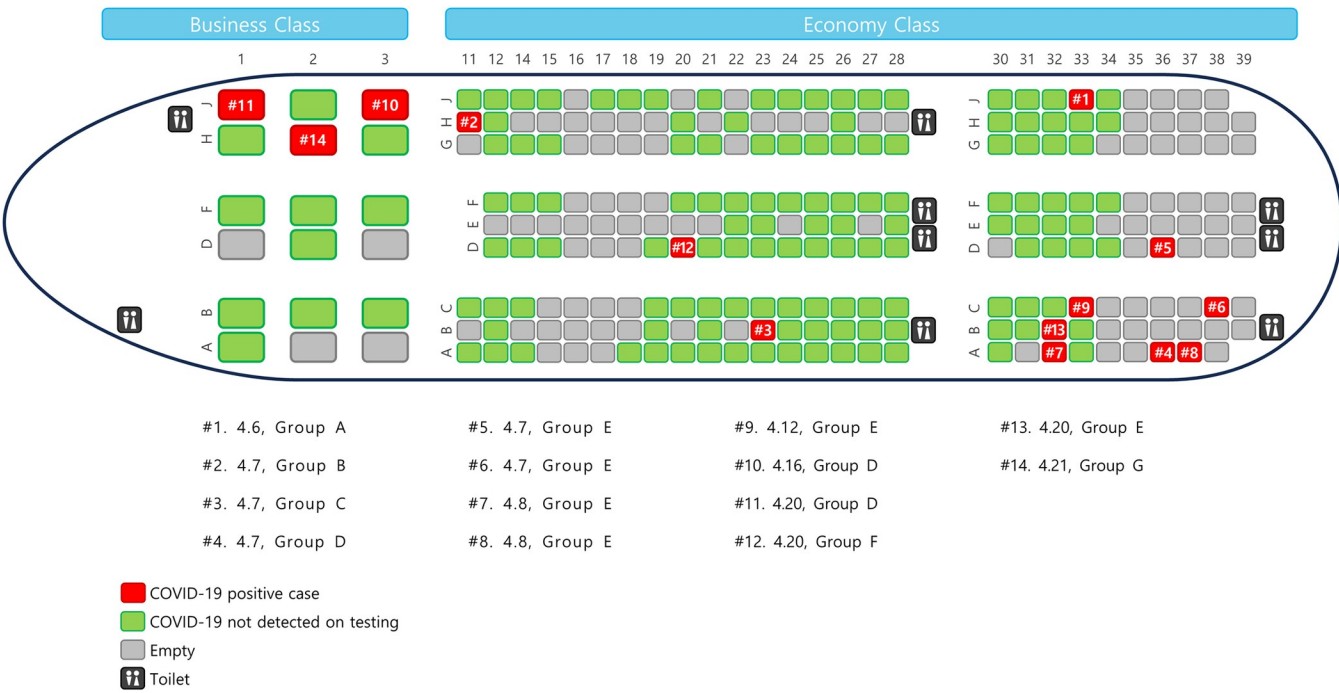

**Fig 1. Airplane seating chart and passenger seats of study subjects.**

**Classification criteria for close contacts.** A total of 256 passenger who could board the aircraft were analyzed, with 18 business class seats and 238 economy class seats. The business class seat had a width of 22 inches and a pitch of 74 inches. The economy class seat had a width of 17 inches and a pitch of 33 inches. Accordingly, the classification of in-flight close contacts in 3 rows and 2 rows was designated as a total of 7 rows and 5 rows, including corresponding rows of confirmed patients. In the case of a business class seat within 1 meter, a close contact within 1 meter was designated as 2 seats left and right and 1 seat front and back. A 1-meter close contact of an economy class seat was designated as 2 seats front and back, left and right. Classification of seats and close contacts of 14 confirmed patients in this study is shown in Fig 1.

The number following the patient index refers to the date of diagnosis and the group refers to the same occupation.

## Ethics approval

The Institutional Review Board of the KDCA approved the protocol. Consent was waived by the ethics committee(KDCA-2023-09-02). This study was collecting information through a prior notice of epidemiological investigation, and is a study using passenger reservation data in accordance with the Quarantine Act, in accordance with Article 16, Paragraph 3 of the Act on Bioethics and Safety. This study was exempted from written consent by the institutional committee because there was no reason to assume that the research subjects would refuse consent, and the risk to the research subjects was extremely low even if consent was waived.

## Data analysis

A descriptive statistical analysis was conducted to analyze the gender, nationality, age, clinical symptoms, COVID-19 variants, COVID-19 genotype, occupational information, and infection rate in the aircraft. All analyses were performed using IBM SPSS ver. 22.0 (IBM Corp.).

## Results

General characteristics and clinical symptom characteristics of COVID-19 confirmed patients are shown in Table 1 and S1 Dataset. There were 14 (100%) males. The nationality was Korean (100%). There were no people under thirty years of age. There were 5 (35.7%) people aged between 30 and 39, 3 (21.4%) aged between 40 and 49, 3 (21.4%) aged between 50 and 59, and 3 (21.4%) aged over 60 years.

When COVID-19 was confirmed, 5 (35.7%) patients showed clinical symptoms with COVID-19 confirmed during entry or quarantine. However, 9 (64.3%) patients were asymptomatic. They were confirmed with COVID-19 on the 1st day of entry or the day before the quarantine was released. As a result of the COVID-19 virus variant test of confirmed patients, 3 (21.4%) were identified as delta virus and 4 (28.6%) were identified as kappa virus. Twelve (85.7%) patients had a G-type and 2 (14.3%) patients had a GH-type. Regarding the period from the date of entry to the date of confirmation, 9 (64.3%) patients were confirmed within 1 to 7 days and 5 (35.71%) patients were confirmed within 8 to 14 days. No patients were confirmed after 14 days from the date of entry. As a result of an epidemiological investigation of 14 patients, 5 patients had no connection between occupations. However, 3 (21.4%) patients had the same workplace in one company and 6 (42.9%) patients had the same workplace in another company. It was found that the three confirmed patients in one company had no close contact or epidemiological relevance in India. The six confirmed patients who had the same workplace in another company had the first confirmed patient at the same workplace two days

**Table 1. General characteristics of 14 patients with COVID-19 confirmed.** (N = 14).

| Variables | Categories | n | % |
|---|---|---|---|
| Gender | Male | 14 | 100.0 |
| | Female | 0 | 0 |
| Nationality | Korean | 14 | 100.0 |
| | Others | 0 | 0 |
| Age range (years) | 30~39 | 5 | 35.7 |
| | 40~49 | 3 | 21.4 |
| | 50~59 | 3 | 21.4 |
| | ≥60 | 3 | 21.4 |
| Symptoms | Yes | 5 | 35.7 |
| | No | 9 | 64.3 |
| Variation status | Delta | 3 | 21.4 |
| | Kappa | 4 | 28.6 |
| | Other variation | 7 | 50.0 |
| Genetic type | G type | 12 | 85.7 |
| | GH type | 2 | 14.3 |
| Entry ~ confirmed period (days) | 1~7 | 9 | 64.3 |
| | 8~14 | 5 | 35.7 |
| Occupational group | A | 1 | 7.1 |
| | B | 1 | 7.1 |
| | C | 1 | 7.1 |
| | D | 3 | 21.4 |
| | E | 6 | 42.9 |
| | F | 1 | 7.1 |
| | G | 1 | 7.1 |

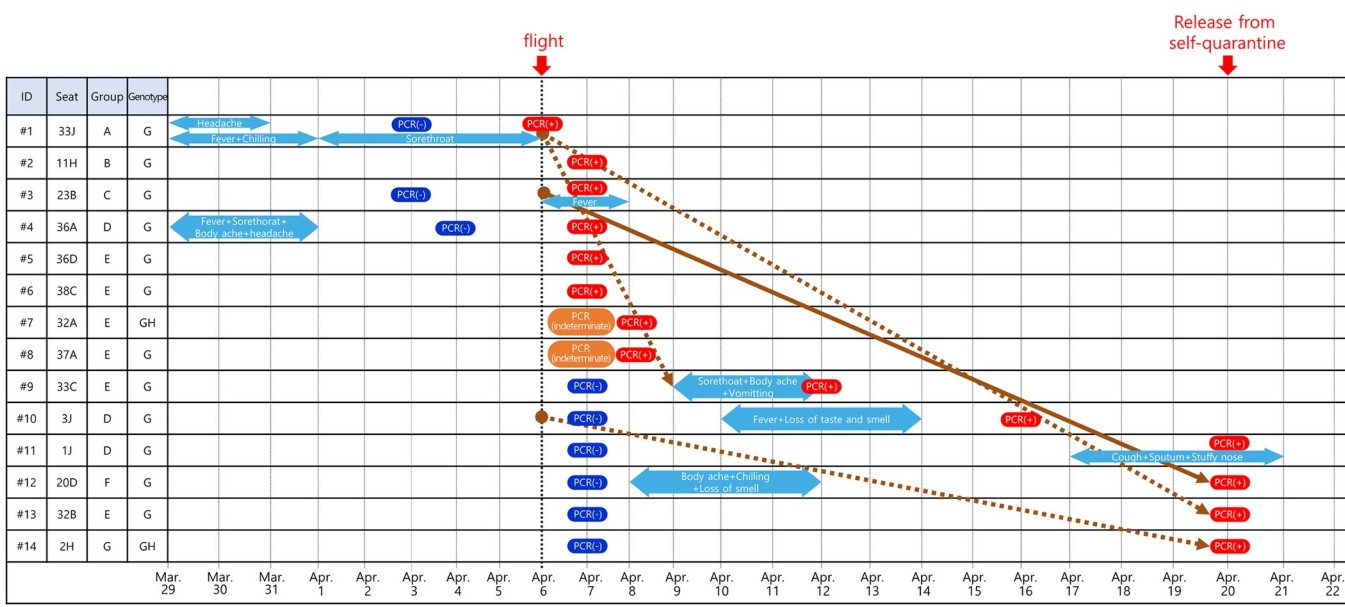

**Fig 2. Date of symptom onset and PCR confirmation of patients with COVID-19.**

before entering the country (April 4, 2023). There were close contacts between employees due to the use of one room for two people in the dormitory and the use of common spaces (toilets and kitchens in the company).

The occurrence date of clinical symptoms and PCR confirmation date of the 14 confirmed patients are shown in Fig 2. Excluding epidemiological connection between the incubation period (1–14 days) and the local close contact, a total of 4 confirmed patients were estimated to be in-flight transmission infections among close contacts within 3 rows. However, as a result of genetic group analysis, the index case (#10) of the #14 confirmed patient was identified as the G group, whereas the #14 confirmed patient was identified as the GH group. Regarding variants, which were smaller categories than genomes, the #1 confirmed patient was identified as the Kappa type and the #4 confirmed patient was identified as Delta type while #9 and #13 were identified as other variables. As a result, infections caused by in-flight transmission were excluded. There was one confirmed patient (#12) who was analyzed to have been infected through close contact in the flight, with the index patient analyzed to be #3. The incubation period for the confirmed patient, who was presumed to have close contact with the index patient during the flight, was 2 days (from the date of exposure to the date of symptom onset). The distance of the confirmed patient from the index patient was more than 1 meter and within 3 rows, including 2 rows.

The periods from the symptoms onset to admission are marked in blue; the incubation period and close contacts but have different genotypes are marked by dot brown arrows; close contacts with the same genotype is marked by solid brown arrow.

Among close contacts of 14 confirmed patients, close contacts with confirmed epidemiological relevance were excluded. One confirmed patient who had COVID-19 due to the possibility of in-flight transmission was used as the index case. Criteria for close contact were 3 rows, 2 rows, 1 meter (2 rows front, back, left, and right). Results of classification and analysis of each infection are shown in Table 2.

**Table 2. Relative risk by distance of confirmed patients with possible in-flight transmission.**

| #3 (23B) | Index case (Within) | Total | Uninfected | Infected | SAR (%) |
|---|---|---|---|---|---|
| | 3row | 50 | 49 | 1 | 2.0 |
| | 2row | 36 | 35 | 1 | 2.8 |
| | 1meter | 12 | 12 | 0 | 0 |

*SAR, Secondary attack rate.

## Discussion

This study was conducted to confirm the possibility of in-flight transmission infection due to close contact in the cabin and to determine whether the risk of infection could decrease by distance from confirmed patients. Ultimately, the purpose of this study is to determine whether in-flight IPC is effective in preventing in-flight infection during the COVID-19 pandemic. The flight analyzed in this study had 180 people on an aircraft with 238 people on board. The flight share was 75.6%. Seating arrangement was 3-3-3 seat Thus, there was a high risk of infection due to close contact in the cabin. In addition, a previous study has reported that COVID-19 delta type has a higher transmission power than existing alpha mutations [18,19]. Seven of the 14 confirmed patients had COVID-19 mutated (four kappa types and three delta types). Although the risk of transmission was high, only one confirmed patient was suspected of in-flight transmission infection due to in-flight close contact. When the date of entry was set as the date of exposure, the confirmed patient showed muscle pain, chills, and decreased sense of smell after 2 days of exposure, similar to the previously revealed COVID-19 incubation period (3–7 days, average 5 days) and clinical symptoms [20–23]. When the variant of the COVID-19 genetic group was analyzed, it was consistent with the index patient. Thus, infection due to in-flight transmission by close contact was suspected. Despite a high possibility of transmission on the flight, only one patient was confirmed due to the possibility of in-flight transmission by close contact, similar to previous studies showing no confirmed patient due to in-flight transmission when infection control and personal hygiene were conducted [11,12,14,15]. According to the epidemiological investigation conducted in this study, the airline crew disinfected before and after the flight to control COVID-19, wore passenger masks, adjusted meal times, wore crew personal protective equipment (gloves, masks, face shields, aprons), and disinfected toilets frequently during the flight. Confirmed patients also followed infection control rules by wearing masks and refraining from unnecessary toilets, which might have reduced the possibility of in-flight close contact.

When calculating the infection rate by distance of confirmed patients with in-flight close contact, it was found that the infection rate was higher within 2 rows than in 3 rows. However, infection did not occur within 1 meter. This seems to be similar to previous studies showing that the closer the distance, the higher the possibility of infection [6,7,14,24]. However, the fact that droplet infection did not occur within 1 meter, the standard of social distance, was thought to be related to the use of aerodynamics and common spaces in the cabin. The air in the cabin flows from the top of the seat to the bottom and exits the floor, creating a kind of air curtain that blocks horizontal airflow. Some of the air that has flowed to the floor mixes with external air and enters the cabin again. In this process, more than 99% of the virus in the air passes through the HEPA filter and enters the cabin [5,25]. Therefore, there were no confirmed patients within the social distance which was conducted to block the spread of general droplet infections. Confirmed case only occurred in the same row. Results of this study, where confirmed infections by COVID-19 occurred within two rows, could be presumed to be

infections caused by the use of common spaces in toilets. However, as a result of the airline's confirmation, toilet disinfection was frequently carried out and transmission by common space is limited for long-distance flights [8]. This study was conducted when delta variant was spreading in India. Infections caused by an unspecified number of local or airport before and after boarding cannot be excluded. The low probability of infection due to in-flight contact is similar to results of previous studies showing that symptom screening before boarding, self-quarantine after boarding, and infection prevention and control during boarding were effective in reducing infections caused by in-flight transmission [6,9,11,12,14,15]. The COVID-19 domestic quarantine measure for overseas entrants, negative confirmation before entry, in addition to being classified as close contacts of two rows front and back, including self-quarantine and confirmed patient rows after entering the country, might have played a role. This suggests that autonomous infection control by passengers and airlines is essential to prevent in-flight infections [26,27].

Risk factors for in-flight infection in early stages of COVID-19 and prior to the pandemic situation have been previously identified [3,7,24]. Due to possible in-flight transmission, infection control is needed with strong quarantine to bring home citizens [12,28,29]. This study is significant in that it shows that autonomous infection control of airlines and passengers on board is essential along with national quarantine measures to prevent the inflow and spread of infectious diseases before and after boarding.

Finally, there is a limitation in generalizing results of this study since it only determined in-flight infection transmission on one flight. Is necessary to study quarantine measures to reduce in-flight infection according to spreading route of infectious disease in the future.

## Conclusions

By confirming the possibility of in-flight transmission during flights with a large number of infectious disease patients and the effectiveness of infection control methods on those flights, domestic quarantine management and infection control methods for infectious diseases that may occur during future travel are discussed.

## Supporting information

**S1 Dataset. Raw data of flight passengers** https://doi.org/10.6084/m9.figshare.26406430.v1. The total number of flight passengers analyzed in this study was 190, including 10 flight attendants and 180 passengers. In this study, data were collected by adding a period of 14 days from the date of the last confirmed case of COVID-19.
(XLSX)

## Author Contributions

**Conceptualization:** Jiyun Park.

**Data curation:** Jiyun Park.

**Formal analysis:** Jiyun Park.

**Funding acquisition:** Gye jeong Yeom.

**Investigation:** Jiyun Park.

**Methodology:** Jiyun Park.

**Project administration:** Jiyun Park.

**Resources:** Jiyun Park.

**Software:** Jiyun Park.

**Supervision:** Gye jeong Yeom.

**Validation:** Gye jeong Yeom.

**Visualization:** Gye jeong Yeom.

**Writing – original draft:** Jiyun Park.

**Writing – review & editing:** Gye jeong Yeom.

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
