## [Decision Letter · Decision Letter 0]

3 Jul 2024

PONE-D-24-16387Risk of COVID-19 transmission on long-haul flights: during the COVID-19 pandemicPLOS ONE

Dear Dr. Yeom,

Thank you for submitting your manuscript to PLOS ONE. After careful consideration, we feel that it has merit but does not fully meet PLOS ONE’s publication criteria as it currently stands. Therefore, we invite you to submit a revised version of the manuscript that addresses the points raised during the review process.

We look forward to receiving your revised manuscript.

Kind regards,

Poowin Bunyavejchewin

Academic Editor

PLOS ONE

Reviewers' comments:

Reviewer's Responses to Questions

**Comments to the Author**

1. Is the manuscript technically sound, and do the data support the conclusions?

Reviewer #1: Yes

Reviewer #2: Yes

2. Has the statistical analysis been performed appropriately and rigorously? 

Reviewer #1: Yes

Reviewer #2: I Don't Know

3. Have the authors made all data underlying the findings in their manuscript fully available?

Reviewer #1: Yes

Reviewer #2: Yes

4. Is the manuscript presented in an intelligible fashion and written in standard English?

Reviewer #1: Yes

Reviewer #2: Yes

5. Review Comments to the Author

Reviewer #1: With the COVID-19 pandemic being one of the most important medically relevant incidents in our recent history, this manuscript explains how the disease was transmitted via air travel (specifically between India and Korea), and also explains in detail how efficient IPC (infection prevention and control) methods need to be implemented to prevent such international catastrophes. Overall the manuscript was well written, detailed and all the data was presented in an easy to understand format.

Reviewer #2: Thank you for allowing me to review your valuable research article.

I think the research topic is very interesting and will contribute to the development of the academic field.

There are two things I would like to ask of you.

1) Please refine the abstract a bit more.

2) There are some awkward English expressions that are common among researchers whose native language is not English. If there is a more refined and clear expression, please reflect it.

6. PLOS authors have the option to publish the peer review history of their article (what does this mean?). If published, this will include your full peer review and any attached files.

Reviewer #1: No

Reviewer #2: No

---

## [Author Response · Author response to Decision Letter 0]

30 Jul 2024

Dear editor and reviewers

We sincerely thank you for pointing out the improvement of the thesis carefully.

We did our best to fix it. Thank you a lot.

---

## [Editor Report · Decision Letter 1]

6 Aug 2024

Risk of COVID-19 transmission on long-haul flights: during the COVID-19 pandemic

PONE-D-24-16387R1

Dear Dr. Yeom,

We’re pleased to inform you that your manuscript has been judged scientifically suitable for publication and will be formally accepted for publication once it meets all outstanding technical requirements.

Kind regards,

Poowin Bunyavejchewin

Academic Editor

PLOS ONE
---

## [Editor Report · Acceptance letter]

8 Aug 2024

PONE-D-24-16387R1 

PLOS ONE

Dear Dr. Yeom, 

I'm pleased to inform you that your manuscript has been deemed suitable for publication in PLOS ONE. Congratulations! Your manuscript is now being handed over to our production team.

Kind regards, 

on behalf of

Mr. Poowin Bunyavejchewin 

Academic Editor

PLOS ONE